# Homochiral antiferromagnetic merons, antimerons and bimerons realized in synthetic antiferromagnets

Mona Bhukta [1], Takaaki Dohi [1,2] ✉, Venkata Krishna Bharadwaj[1], Ricardo Zarzuela[1], Maria-Andromachi Syskaki [1,3], Michael Foerster [4], Miguel Angel Niño [4], Jairo Sinova[1], Robert Frömter [1] ✉ & Mathias Kläui[1] ✉

The ever-growing demand for device miniaturization and energy efficiency in data storage and computing technology has prompted a shift towards antiferromagnetic topological spin textures as information carriers. This shift is primarily owing to their negligible stray fields, leading to higher possible device density and potentially ultrafast dynamics. We realize in this work such chiral in-plane topological antiferromagnetic spin textures namely merons, antimerons, and bimerons in synthetic antiferromagnets by concurrently engineering the effective perpendicular magnetic anisotropy, the interlayer exchange coupling, and the magnetic compensation ratio. We demonstrate multimodal vector imaging of the three-dimensional Néel order parameter, revealing the topology of those spin textures and a globally well-defined chirality, which is a crucial requirement for controlled current-induced dynamics. Our analysis reveals that the interplay between interlayer exchange and interlayer magnetic dipolar interactions plays a key role to significantly reduce the critical strength of the Dzyaloshinskii-Moriya interaction required to stabilize topological spin textures, such as antiferromagnetic merons, in synthetic antiferromagnets, making them a promising platform for next-generation spintronics applications.

Recent years have witnessed an increasing interest in chiral magnetic topological spin textures such as skyrmions[1,2], biskyrmions[3–5], hopfions[6,7], chiral bobbers[8,9], and skyrmionic cocoons[10] due to their potential use as information carriers for high-density data storage and (un)conventional computing[11–15]. These chiral topological spin textures are primarily stabilized by the interplay between the Dzyaloshinskii-Moriya interaction (DMI)[16,17], the perpendicular magnetic anisotropy (PMA), and the dipolar interaction. The orientation of the DMI vector, which is determined by the way the inversion symmetry is broken, sets the preferential stabilization of diverse skyrmion types, such as

Bloch[1,2], Néel[18,19], or antiskyrmions[20]. For instance, in magnetic multilayers, the combination of intrinsic inversion symmetry breaking, and strong spin-orbit coupling provided by heavy atoms at the interface of a ferromagnetic/heavy-metal heterostructure creates an ideal platform for providing interfacial DMI[21], which in turn stabilizes Néel-type skyrmions. In magnetic thin films, these Néel skyrmions exhibit significant topological robustness[22] and are amenable to electrical control[18,23–25], but also entail disadvantages such as the skyrmion Hall effect[18,24,25]. The growing demand for high-speed, low-power technologies has therefore boosted the search for more complex topological spin textures

[1]Institute of Physics, Johannes Gutenberg-University Mainz, 55099 Mainz, Germany. [2]Laboratory for Nanoelectronics and Spintronics, Research Institute of Electrical Communication, Tohoku University, 2-1-1 Katahira, Aoba, Sendai 980-8577, Japan. [3]Singulus Technologies AG, Hanauer Landstrasse 107, 63796 Kahl am Main, Germany. [4]ALBA Synchrotron Light Facility, 08290 Cerdanyola del Vallés, Barcelona, Spain. ✉e-mail: tdohi@tohoku.ac.jp; froemter@uni-mainz.de; klaeui@uni-mainz.de

beyond the skyrmion paradigm. Topological spin textures in in-plane magnets, such as merons[26], antimerons[27], and bimerons[28–31] have recently been explored, by virtue of their richer (current-induced) dynamics compared to skyrmions[32] and the stackability, a property that allows for denser quasi-one-dimensional racetracks in three dimensions, resulting in higher storage density. Bimerons are robust topological textures that are homeomorphic to skyrmions and can be visualized as the combination of two half-skyrmions (merons). When arranged in different in-plane directions, merons offer additional degrees of freedom compared to conventional skyrmions, making them an important focus in fundamental quasi-particle research as well as topology-based computing approaches. Despite the advantage of having the ability to stabilize pure homochiral spin textures, ferromagnetic (FM) topological spin textures suffer from limitations in terms of scalability with respect to sufficient thermal stability[33], stackability due to long-range magnetic dipolar interactions[34], and gyrotropic forces resulting from their net intrinsic spin angular momentum.

Antiferromagnetic (AFM) systems can naturally overcome these inherent limitations of FM textures, due to their compensated spin angular momentum and negligible stray fields[35–42]. While one could envisage using single-crystalline antiferromagnets, the inherent technological advantages are challenged by the difficulty of stabilizing pure homochiral spin textures. These challenges stem from the absence of significant Lifshitz invariants, resulting in the observed spin structures having random chirality[43–46]. So far, in antiferromagnets the observation of in-plane (IP) topological spin textures such as bimerons has been limited to observing their helicity, in spite of recent advances in their creation via sophisticated protocols[43–46]. The anticipated dynamics of topological spin textures in the presence of spin-orbit torques (SOTs) are heavily influenced by their helicity, leading to Bloch-type and Néel-type structures moving perpendicular and along the direction of SOT, respectively[25]. However, the lack of homochirality of spin textures in native single-crystalline antiferromagnets limits their use for controlled dynamics of spin textures and thus prevents their utilization in future spintronics devices.

An ideal system to explore and manipulate both structural and dynamical properties of (IP) topological spin textures are synthetic antiferromagnetic (SyAFM) platforms[36–38,47,48]. They consist of a multi-layered heterostructure made of FM thin films separated by non-magnetic metallic spacers and antiferromagnetically coupled via the interlayer exchange interaction[49,50]. By tailoring the amount of compensation they can exhibit an arbitrarily small magnetic moment and, therefore, combine the most interesting features of both FM and AFM scenarios: minimal stray fields, the ability to stabilize homochiral spin textures, and the potential for ultrafast spin dynamics; all in a device-compatible easy to fabricate polycrystalline multilayer setting. A precondition to assess the topological nature of such spin textures is to be able to measure their chirality. In this regard, synthetic antiferromagnets offer the advantage to employ the advanced surface- or element-sensitive imaging methods available for ferromagnet. AFM IP topological spin textures can be formed locally during the magnetization reversal process[51]. However, spontaneously formed stable homochiral spin textures on a global scale have hitherto not been reported.

In this article, we employ multimodal vector imaging of the three-dimensional (3D) staggered magnetization to demonstrate the successful stabilization of all members of the class of IP AFM topological spin textures emerging in a specifically designed layered synthetic antiferromagnet, namely merons, antimerons, and bimerons at zero magnetic fields. Our experiments combine magnetic force microscopy (MFM), scanning electron microscopy with polarization analysis (SEMPA), and element-specific photoemission electron microscopy using the X-ray magnetic circular dichroism (XMCD-PEEM), which enable us to identify spin textures possessing enhanced stability, classified by integer, nonzero topological invariants, as well as

topologically trivial ones. We find that in the vicinity of the spin-reorientation transition (SRT), where the effective anisotropy vanishes, the SyAFM platform can host homochiral AFM merons, as determined from their helicity and core polarity. Furthermore, their helicity can be easily tailored by the degree of magnetic compensation of the synthetic antiferromagnet, indicating that interlayer dipolar interactions play a significant role in the stabilization of these spin textures. Our micromagnetic and analytical calculations fully explain the experimental observations, elucidate the mechanism for the stabilization of AFM topological textures in synthetic antiferromagnets, and describe the corresponding phase diagram of their stability. Our findings provide crucial insights into the formation and stability of homochiral IP AFM topological textures, which pave the way towards better scalable soliton-based technologies beyond the skyrmion paradigm.

## Results

### Tuning the magnetic properties to stabilize AFM (anti)merons

Choosing FM materials with low pinning, negligible PMA, and finite DMI, as well as a strong AFM coupling between adjacent FM layers is a key requirement for the stabilization of (anti)merons and bimerons in SyAFM platforms, as devised later in the discussion section. We have therefore optimized a Pt/CoFeB/Ir-based multilayer synthetic antiferromagnet (see Methods for details). As depicted in Fig. 1a, a heterostructure consisting of multiple repetitions (14 times) of the single bilayer synthetic antiferromagnet has been prepared to achieve two main objectives: (1) reduce the thermal diffusion of (anti)merons across the low-pinning synthetic antiferromagnet, and (2) obtain high saturation fields. The two FM sublattices of the synthetic antiferromagnet are denoted by $FM_A$ and $FM_B$, respectively, with their saturation magnetization represented by $M_{s,A}$ and $M_{s,B}$. The FM films consist of the bilayer $Fe_{0.6}Co_{0.2}B_{0.2}$(FCB)/$Co_{0.6}Fe_{0.2}B_{0.2}$(CFB) and are sandwiched between the nonmagnetic spacers Pt and Ir. This breaks the mirror symmetry of the heterostructure, thus providing a finite DMI ($D$). The topmost Pt layer serves as capping for the material stack to prevent oxidation over time and during post-processing (patterning). The thickness $d_{Ir} = 0.400$ nm of the Ir layer is chosen so as to maximize the AFM interlayer exchange between the FM layers. The CFB layer induces PMA ($K_u$) at the interface with the heavy metal, whereas the thickness ratio to the FCB layer is used to control the magnetic dipolar anisotropy $K_d = -\frac{1}{2}\mu_0 M_s^2$. We have kept the FM layers as thin as possible to maximize the interlayer exchange coupling ($d_{FM} = 0.900$ nm) as well as optimized the ratio between the CFB ($x$ nm) and FCB ($0.9 - x$ nm) thicknesses to be in the vicinity of the spin reorientation transition (namely, to obtain a vanishing effective anisotropy $K_{eff} = K_u + K_d$).

Figure 1b, c depict the $M(H)$ loops of the stacks #1 ($x = 0.7$) and #2a ($x = 0.2$), respectively, where the red (blue) curve represents the magnetic hysteresis loop for an IP (out-of-plane) configuration of the external magnetic field. For $x = 0.7$, the SyAFM exhibits a positive effective anisotropy (as deduced from the appearance of the blue shaded region in Fig. 1b), resulting in out-of-plane (OOP) multidomain states at room temperature and zero magnetic field. In Fig. 1c, for $x = 0.2$, red and blue hysteresis loops coincide, indicating $K_{eff} \simeq 0$. Furthermore, the zero remanence makes the stack #2a a potential candidate for hosting (bi)merons, as the formation of a multi-domain magnetic ground state is expected for this stack at zero field. Stack #2b is intentionally engineered such that the FM layers have a small (normalized) uncompensated magnetization, denoted as $m_{uncomp} = \frac{|M_{s,A} - M_{s,B}|}{M_{s,A} + M_{s,B}} = 5\%$. This design choice enables the detection of the OOP spin components of the meron spin textures through MFM imaging, and the corresponding hysteresis loop is shown in Fig. 1d. The effect of the magnetic compensation in synthetic antiferromagnets on the formation of merons has been studied via stack #3, and the hysteresis loop is shown in Fig. 1e, indicating $m_{uncomp}$ to be 20%. In this case, FM layers A and B are made of $Co_{0.8}B_{0.2}$ (CB) and FCB,

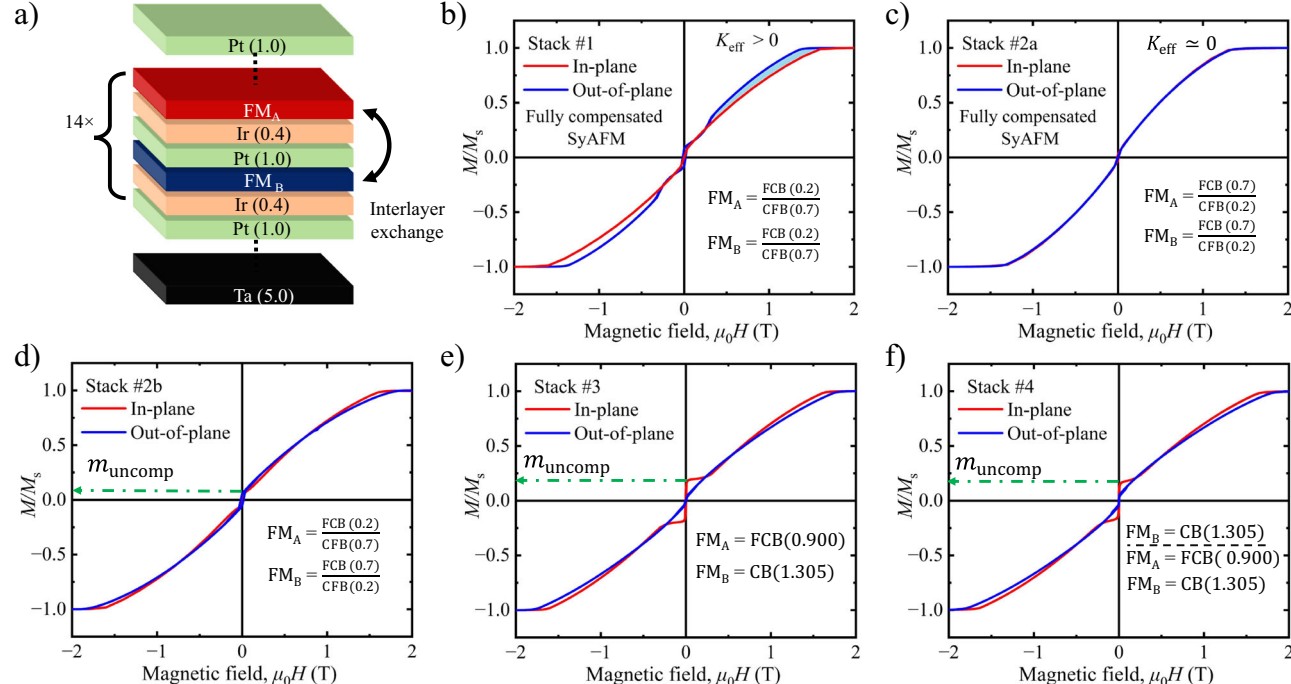

**Fig. 1 | Material stack and magnetic properties of the SyAFM. a** Multilayer structure for the SyAFM system, where $FM_A$ and $FM_B$ denote the AFM-coupled FM sublattices. The numbers in parenthesis give the thickness in nm (**b–f**). The thicknesses of the FM bilayers are indicated on the respective graphs. OOP (blue curve) and IP (red curve) hysteresis loops measured by means of SQUID magnetometry for the stack (**b**) #1 (fully compensated; $K_{eff} > 0$), (**c**) #2a (fully compensated; $K_{eff} \simeq 0$), (**d**) #2b (95% compensated; $K_{eff} \simeq 0$), (**e**) #3 (80 % compensated; $K_{eff} \simeq 0$), and (**f**) #4 (identical to stack #3, with one additional $FM_B$ unit on the top). The presence of the blue-shaded region in (**b**) indicates positive $K_{eff}$, whereas the rest of the SyAFM stacks shows an overlap between the IP and OOP $M(H)$ curves, which indicates vanishing $K_{eff}$. The green dashed line in (**d–f**) represents the magnetic uncompensation present in the synthetic antiferromagnet.

respectively, with thicknesses of $d_{FM_A} = 1.305$ nm and $d_{FM_B} = 0.900$ nm. Stack #4 is on purpose designed to harness the element specificity of XMCD-PEEM, thus enabling the visualization of the antiferromagnetic coupling between the FM layers. It contains one additional $FM_B$ layer on the top of stack #3 and its magnetic hysteresis loop is shown in Fig. 1f.

## Multimodal 3D-vector imaging of merons/antimerons in the SyAFM system

Next, we reveal the topological spin textures present in the synthetic antiferromagnet. For the full vector reconstruction of the Néel order parameter $\mathbf{L} = \mathbf{M}_A - \mathbf{M}_B$, imaging of both IP and OOP spin components of the meron spin structures is required. The slightly uncompensated stack #2b produces small stray fields and thus, by using the combination of SEMPA and MFM imaging techniques on the very same area, allows us to reconstruct the Néel order describing the topological textures in these SyAFM platforms. Figure 2a, b show a SEMPA image (IP spin components) and a MFM image (OOP spin component) taken in exactly the same area of stack #2b at room temperature and zero magnetic field. The observed topological spin textures were created by applying a damped oscillating OOP magnetic field prior to imaging. Owing to the profound surface sensitivity of the SEMPA technique[52], the detected contrasts in Fig. 2a represent exclusively the IP magnetization vector of the uppermost $FM_A$ layer, thereby providing insight into the IP direction of the Néel order of the synthetic antiferromagnet. By analyzing the IP color-coded images shown in Fig. 2a, we ascertain the winding number $w$ that is connected with the meron spin textures. This is achieved by observing the rotational behavior of the IP magnetization direction. $w$ provides a measure of wrapping of the Néel order around the unit sphere and it reads $w = \pm 1$ for skyrmions/bimerons, whereas it becomes $w = \frac{1}{2}$ for a meron and $w = -\frac{1}{2}$ for an antimeron. Moreover, the analysis of the in-plane rotation direction of

these structures from Fig. 2a provides the precise helicity $\gamma$ of these meron spin structures. $\gamma$ is given, akin to skyrmions[11], by the angle between the IP projection of the Néel order and the radial direction. The observation reveals the presence of three distinct spin textures: (1) Néel merons having $\gamma = \pi$, indicated by black dotted circles (2) Néel merons having $\gamma = 0$, indicated by white double circles and (3) the spin texture of antimerons, indicated by double black dotted circles.

In this SyAFM stack, the saturation magnetization of layer A, $(M_{s,A})$, surpasses that of layer B, $(M_{s,B})$. As a result, the MFM measurement primarily detects the stray field from the A layers, similar to SEMPA, thus indicating the OOP component of the Néel order parametrizing the SyAFM system. Consequently, through the visualization of the OOP core polarities of the spin textures identified in Fig. 2a, we reveal the full 3D Néel vector of the synthetic antiferromagnet. This allows us to determine their topological charge $Q$, which can be cast as the product of the winding number and the core polarity, namely $Q = w \cdot L_z|_{core}$, the core polarity being defined as the $z$ component of the Néel order at the texture core.

The white and dark brown contrasts in Fig. 2b show the downward and upward core polarity, respectively. Furthermore, black and white circles in both images correspond to $Q = -\frac{1}{2}$ and $Q = \frac{1}{2}$, respectively. The analysis of both images reveals the topological nature of the meron spin textures: black dotted circles represent Néel-type merons having $Q = -\frac{1}{2}$ with core polarity pointing downward ($L_z = -1$) as indicated in from Fig. 2d. White double circles correspond to merons of $Q = \frac{1}{2}$ having upward core polarity, as described in Fig. 2c. We note that for merons the helicities $\gamma = 0, \pi$ always correspond to the core polarities $L_z = 1$ and $-1$, respectively, which indicates the presence of homochiral merons in the system. Double black dotted circles mark antimerons with topological charge $Q = -\frac{1}{2}$, see Fig. 2e. Changes in $\gamma$ do not affect the topological charge of the antimeron as they only lead to a geometrical rotation of its IP spin components. Hence, we

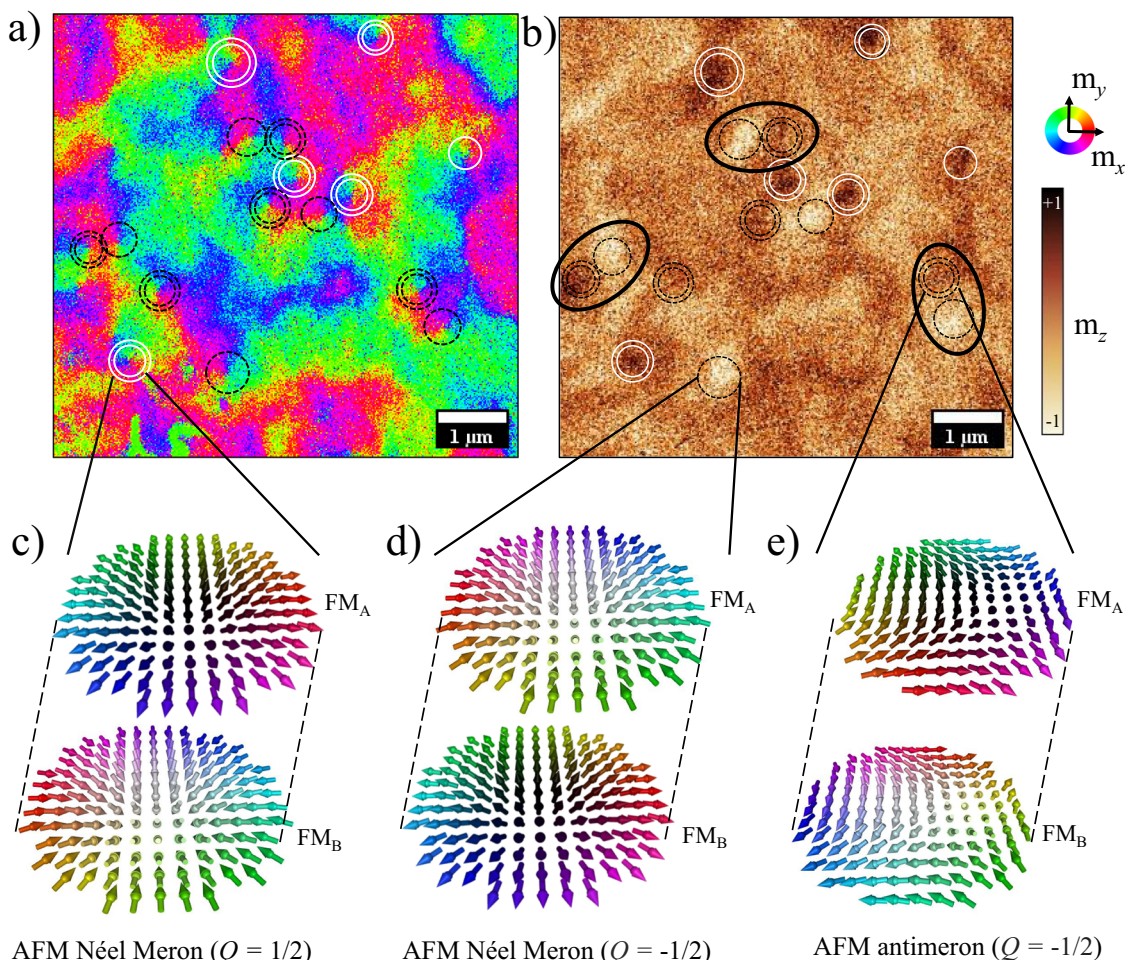

**Fig. 2 | Imaging the Néel order parameter of (anti)merons and bimerons in synthetic antiferromagnets. a** SEMPA image showing the IP spin components of the topmost $FM_A$ layer, or, equivalently, the IP component of the Néel order of the meron textures in the stack #2b. **b** MFM image showing the OOP spin component of the $FM_A$ layers, that is the OOP component of the Néel order of the meron structure in the same area. Dark brown and white MFM contrasts indicate the upward and the downward direction, respectively. The color map for the SEMPA image is shown on the right side. Black dotted circles represent merons of helicity $\gamma = \pi$, whereas double black dotted circles indicate antimerons, both with $Q = -\frac{1}{2}$. White circles represent merons having an arbitrary helicity with $Q = \frac{1}{2}$ and white double circles denote merons of helicity $\gamma = 0$ and $Q = \frac{1}{2}$. Two adjacent black circles (single and double) are identified as bimerons with net topological charge $Q = -1$ and are additionally highlighted by ellipses. **c** an AFM Néel meron having $\gamma = 0$ and $Q = \frac{1}{2}$, (**d**) an AFM Néel meron having $\gamma = \pi$ and $Q = -\frac{1}{2}$ and (**e**) an AFM antimeron having $Q = -\frac{1}{2}$.

can consider all of them as topologically equivalent. The combination of a single black circle adjacent to a double one can be identified as a bimeron with $Q = -1$ and marked by an ellipse. AFM coupling between the meron spin textures present in the adjacent FM layers has been confirmed by means of XMCD-PEEM layer-resolved imaging (for details, see Supplementary Section 2). Next, we quantify the spin structures and their statistics. Figure 3a presents typical line profiles of the MFM phase image representing the up and down core polarities. These line profiles were acquired for stack #2b under zero magnetic fields and correspond to the positions specified in the insets. The Gaussian peak functions fitted to the measured points along the line profiles reveal that the profiles from different cores coincide within the resolution limit of the instrument. The full width at half maximum of the Gaussian fit provides a measure of the diameter of the cores[37]. The diameter distribution of the (anti)meron cores, as extracted from the larger MFM image in Supplementary Fig. 4b, is shown in Fig. 3b. We find average diameters of $(232 \pm 50)$ nm and $(220 \pm 60)$ nm for up and down cores, respectively, which are identical within the given uncertainty. When examining the spatial arrangement of the (anti)merons in Fig. 2a, it is apparent that in most cases merons and antimerons are positioned in close proximity to each other. Meron-antimeron

composites can exhibit either a non-zero topological charge, arising when the core polarities are different, or be trivial spin textures with $Q = 0$, occurring when the core polarities are the same. To gain a more comprehensive understanding of the length scale of their interaction, we conducted a statistical analysis of the distance between these meron textures over a larger sample area. In Fig. 3c, three histograms of the next-neighbor distances between different core polarities, as observed in the MFM images, are presented. Up-down pairs exhibit an average separation of $(530 \pm 240)$ nm, which is significantly closer than the separation between two up or two down pairs. This closer proximity strongly suggests the presence of non-zero topological charge in the meron-antimeron pairs and indicates the prevalence of bimerons in the sample. Additionally, this observation implies a different interaction potential with an energy minimum at shorter distance between two different core polarities due to the presence of DMI. The statistical analysis presented in this work considers only the next-nearest neighbor distances between meron spin textures. This choice is based on the fact that the stabilization of merons is attributed to the DMI, as supported by the existence of homochiral Néel merons in the system[53]. Furthermore, as depicted in Fig. 3a, there are instances where (anti)merons occur in high-density clusters. It has been reported that

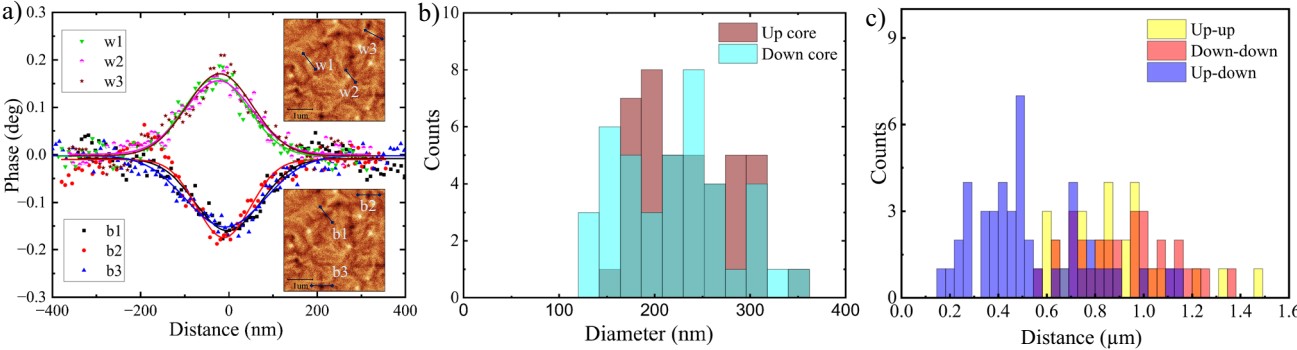

**Fig. 3 | Quantitative analysis of the MFM contrast. a** Line profiles b1, b2, and b3 show the MFM signal of the up core polarity, while the line profiles w1, w2, and w3 represent the MFM signal of the down core polarity. The solid curves correspond to fits of the measured points using Gaussian peak functions. The horizontal axis gives the distance along each corresponding line profile form the meron cores, as depicted in the insets. **b** Distribution of different core diameters obtained from various MFM images. **c** Histogram of the next-neighbor separation between the centers of structures with up-up, down-down, and up-down core polarities from the MFM phase contrast.

as a result of manipulating temperature and magnetic fields, the intermittent emergence of tightly packed spin-texture clusters can be effectively managed[43]. For instance, the (anti)merons can be erased and recreated by utilizing magnetic fields, as explained in detail in Section 3 of the Supplementary.

## Theoretical model

Our experiments show the existence of room-temperature stable bimerons in compensated synthetic antiferromagnets at zero external magnetic fields. Nevertheless, our findings give rise to several intriguing questions. The stabilization mechanism of bimerons in the synthetic antiferromagnet platform needs to be clarified, especially considering that these structures have been previously predicted only in ferromagnetic materials with monoclinic symmetry[32,54]. The stabilization of skyrmions in SyAFM systems relies on the key factors of vanishing anisotropy and the presence of DMI. As bimerons are the in-plane analogues of skyrmions, it remains essential to explore how these parameters influence the stabilization of bimerons in the SyAFM system. To address these intriguing questions, we have developed an analytical model and performed micromagnetic simulations using experimentally extracted parameters, as detailed in the Methods section.

We start with a synthetic antiferromagnet that can be effectively described, irrespective of its magnetic compensation ratio, as a ferrimagnetic platform with magnetic sublattices given by the A and B FM layers, respectively. In the compensated case, the minimal model for the synthetic antiferromagnet contains exchange, DMI, and anisotropy contributions, and thus its total free energy reads

$$\mathcal{E}[\boldsymbol{L}] = \int_{\mathcal{S}} d^2\boldsymbol{r} \left[ \frac{A}{2}(\nabla\boldsymbol{L})^2 + D\boldsymbol{L} \cdot (\tilde{\nabla} \times \boldsymbol{L}) - KL_z^2 \right], \quad (1)$$

where $A$ is the AFM spin stiffness constant and $\mathcal{S}$ denotes the SyAFM surface (see Supplementary Section 5). The effective (easy-axis) anisotropy constant for the Néel order, $K = K_{\mathrm{eff}} - \frac{H_d^2}{2\delta L^2}$, has an additional contribution originating in the competition between the interlayer exchange and interlayer magnetic dipolar interactions. Here, $H_d$ and $\delta$ denote the interlayer stray field and the interlayer exchange constant, respectively. In the vicinity of the SRT point, $K_{\mathrm{eff}} \sim \frac{H_d^2}{2\delta L^2} \ll K_u$, and therefore the interlayer magnetic dipolar field can induce the reorientation (from OOP to IP) of the staggered magnetization describing the synthetic antiferromagnet, which in turn favors the stability of chiral topological spin textures. Thus the formation of meron spin textures in synthetic antiferromagnets results from this subtle interplay between interlayer exchange, interlayer magnetic dipolar

interactions, and interfacial DMI, as well as from the effective anisotropies of the FM layers. We note that FM systems can be tuned close to zero effective anisotropy but lack the interlayer magnetic dipolar field, whereas AFM platforms have a spin-flop contribution to the effective anisotropy but their SRT is usually driven by temperature[43]. The antiferromagnet lacks inversion symmetry due to antiferromagnetic ordering, which significantly reduces the magnitude of the Lifshitz invariants, and therefore it lacks the necessary criteria to stabilize homochiral spin structures.

The phase diagram in Fig. 4a illustrates the presence of distinct ground states within the synthetic antiferromagnet as a function of DMI and effective anisotropy. Here, $2\delta$ is set to $4.4 \times 10^{-4}$ Jm$^{-2}$, matching the interlayer exchange strength of the experiment. The bronze-colored region in the phase diagram represents the helical phase in the $rz$ plane, wherein the ground-state phase exhibits helical magnetic ordering along an arbitrary radial direction $r$. As the synthetic antiferromagnet is tuned away from the SRT point towards an IP configuration (i.e., $K < 0$), the critical value of DMI to stabilize bimerons ($D_c^{\mathrm{IP}}$) increases, since one needs to overcome a larger anisotropy barrier to induce the OOP tilting of the staggered magnetization. The phase boundary between the uniform IP (dark green region) and the helical phases has been calculated analytically (for details, see Methods) and is described parametrically by the curve $D_c^{\mathrm{IP}} = \frac{4}{\pi}\sqrt{A\left[\frac{H_d^2}{2\delta L^2} - K_{\mathrm{eff}}\right]}$. Furthermore, the ground state is the uniform OOP configuration (yellow region) when $K > 0$ and, as the DMI increases above $D_c^{\mathrm{OOP}}$, a phase transition towards the helical state is induced, which is well-known in systems with PMA. The light green color depicts the region where AFM bimerons are stabilized as derived from micromagnetic simulations. The ground-state phases coalesce at the triple point ($D_c$) in the $D - K_{\mathrm{eff}}$ phase diagram, marked with a black star, which represents the critical DMI required to stabilize homochiral AFM bimerons. The expression for ($D_c$) is derived from the condition $K_{\mathrm{eff}} = 0$, resulting in the equation $D_c = \frac{4}{\pi}\sqrt{A\left[\frac{H_d^2}{2\delta L^2}\right]}$ (for details, see Methods). For a strongly AFM-coupled synthetic antiferromagnet (i.e., large $\delta$), in the vicinity of the SRT point (see Supplementary Fig. S7), only a very small DMI is needed to induce the phase transition from the uniform IP configuration to a helical phase (denoted by the green and bronze areas). The low $D_c$ stems from the fact that the only contribution to the effective anisotropy at the SRT point comes from the interlayer magnetic dipolar field. We observe that the critical value $D_c$ is proportional to $\frac{1}{\sqrt{\delta}}$, leading to an increase in $D_c$ for weak AFM interlayer couplings, as shown in Fig. 4b with $1 \times 10^{-5}$ Jm$^{-2}$, and Fig. 4c with $1 \times 10^{-6}$ Jm$^{-2}$. Achieving a larger DMI

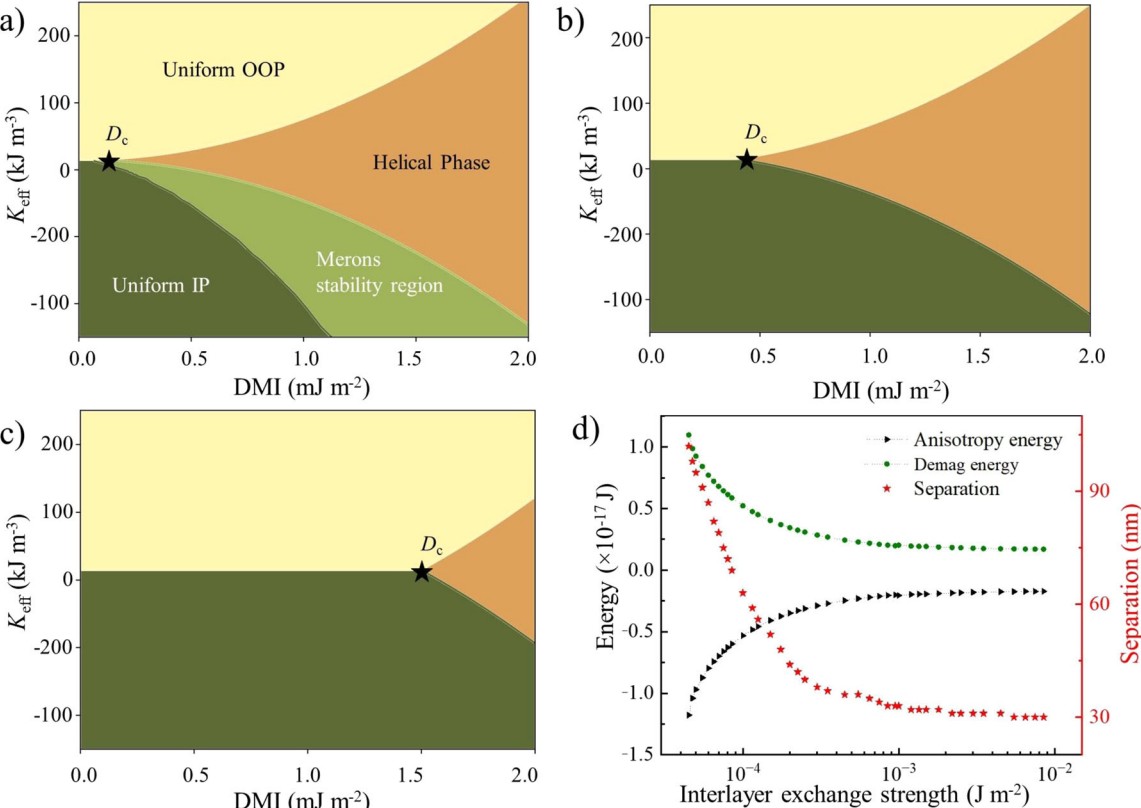

**Fig. 4 | Phase Diagram and micromagnetics of bimerons in synthetic anti-ferromagnets.** $D$ vs $K_{eff}$ phase diagrams corresponding to different values of $2\delta$: (**a**) $2\delta = 4.4 \times 10^{-4}$ Jm$^{-2}$, (**b**) $2\delta = 1 \times 10^{-5}$ Jm$^{-2}$, and (**c**) $2\delta = 1 \times 10^{-6}$ Jm$^{-2}$. The ground-state phases converge at the triple point $D_c$ in the $D - K_{eff}$ phase diagram, indicated by a black star. Dark green, yellow, and bronze colors represent the uniform IP, uniform OOP, and helical ground-state phases in the $rz$ plane, respectively, with $r$ being any

radial direction. The light green color depicts the region where AFM merons are stabilized in micromagnetic simulations. The phase diagrams in (**b**, **c**) illustrate a displacement of the triple point to the right. **d** Evolution of the separation between the two cores of different polarities of a bimeron and its energy terms; anisotropy energy ($E_{ani}$), and demagnetizing energy ($E_{demag}$) as a function of the interlayer exchange strength.

for an easy-plane system can be exceedingly challenging experimentally; consequently, systems with strong AFM interlayer coupling offer a more viable route for stabilizing bimerons.

In Fig. 4d, the red stars represent the separation between the cores of opposite polarities of the meron and antimeron within a bimeron, obtained from micromagnetic simulations using Mumax3 (see Methods for more details). These simulations are performed for $D = 0.16$ mJm$^{-2}$ and $K_{eff} = -0.08$ MJm$^{-3}$ for a fully compensated synthetic antiferromagnet. As the interlayer exchange strength increases, the system's anisotropy energy ($E_{ani}$) decreases, as shown by the black triangular points. Consequently, this destabilizes the OOP magnetization within the bimeron, leading to smaller bimeron sizes and shorter separations between the cores. Additionally, this change causes a decrease in the stray field and demagnetization energy ($E_{demag}$) of the bimeron.

## Tailoring the helicity of (anti)merons in synthetic antiferromagnets

The chirality and helicity of skyrmions are intricately related to their stabilization mechanisms. Néel Skyrmions, primarily stabilized by interfacial DMI, exhibit energy minima for helicities $\gamma = 0$ or $\gamma = \pi$, while Bloch skyrmions, stabilized through dipolar interactions, tend to favor $\gamma = \pi/2$ or $\gamma = 3\pi/2$[11]. Consequently, identifying and manipulating the helicity plays a pivotal role in unraveling the underlying stabilization mechanisms and engineering the SOT-induced dynamics. In their in-plane counterparts, a compelling question arises regarding the effective manipulation of (anti)meron helicity. In this section, we demonstrate the control of the helicity of merons in synthetic

antiferromagnets by tuning the magnetic compensation ratio, which in turn controls the dipolar energy of the system.

Figure 5a shows a SEMPA image indicating the direction of the in-plane Néel order, for the fully compensated synthetic antiferromagnet (stack #2a) in the absence of magnetic fields. We have marked all merons and antimerons find an almost equal proportion of both types of topological spin textures. We elucidate the relevance of the helicity by analyzing its values for the merons through a histogram, as shown in Fig. 5b. This histogram has been aggregated by also considering additional SEMPA images of the stack #2a obtained under comparable conditions. Values of $\gamma = 0, \pi$ are significantly favored in this SyAFM platform, which corresponds to the Néel-type rotation, and therefore confirms that the stabilization mechanism for merons in the compensated case originates from the DMI[32,53,55]. Similarly, Néel bimerons are energetically favorable in the same range of DMI[32]. For comparison, the uncompensated case has been studied in stack #3, which presents an $m_{uncomp} = 20\%$ and thus a small but significant interlayer dipolar field. A SEMPA image of its topmost-layer magnetization is shown in Fig. 5(c) giving the direction of the net IP magnetization. We observe again a nearly equal number of merons and antimerons. Black triangles and black stars represent merons with helicities of $3\pi/2$ and $\pi/2$, respectively. However, for this stack the analysis of the distribution of meron helicities, see Fig. 5d, yields a prevalence of values around $\gamma = \frac{\pi}{2}, \frac{3\pi}{2}$, which indicates a Bloch-type rotation. In the picture of the analytical model developed above, an additional Zeeman-like interaction $-\frac{\Theta}{L^2}H_d L_z$ contributes to the energetics of the SyAFM, which favours the OOP orientation of the Néel order and, therefore, the stability of Bloch-type IP merons at low DMI. Here, $\Theta = M_{s,A}^2 - M_{s,B}^2$

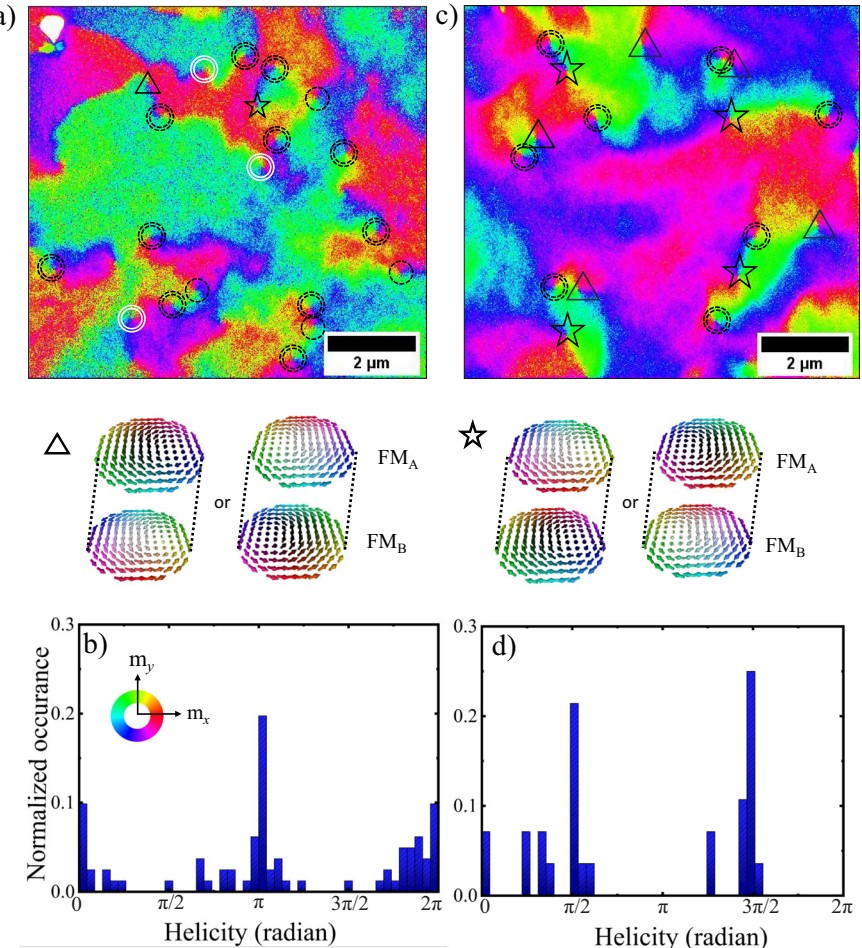

**Fig. 5 | Manipulating the helicity of (anti)merons in synthetic antiferromagnets. a** SEMPA image showing the IP spin components of the meron spin texture in the stack #2a indicating the IP orientation of the staggered magnetization. Black dotted circles represent merons with helicity of $\gamma = \pi$, while double black dotted circles indicate antimerons, and white double circles denote merons with a helicity of $\gamma = 0$. Black triangles and black stars represent merons with helicities of $3\pi/2$ and $\pi/2$, respectively. **b** Distribution of helicities of the meron present in the synthetic antiferromagnet. The abundance of helicity values around 0 and $\pi$ indicates homochiral Néel merons in the stack. **c** SEMPA image showing the IP spin components of the meron spin texture in the (uncompensated) case of stack #3. **d** shows the prevalence of Bloch merons having helicities of $\pi/2$ and $3\pi/2$ in the stack.

parametrizes the amount of magnetic uncompensation in the synthetic antiferromagnet. We conclude that the presence of a small uncompensated magnetization in the synthetic antiferromagnet promotes the stabilization of Bloch-meron textures. Thus by tuning the compensation ratio, we can manipulate the helicity and consequently, one can tune the resulting SOT-induced dynamics.

## Discussion

In conclusion, we have created chiral merons, antimerons, and topologically stabilized bimerons in synthetic antiferromagnets at zero magnetic fields. The direction of the net magnetization and the emergent field created by topology in bimerons are mutually orthogonal[56], a key difference to their OOP counterparts, skyrmions. Thus meron spin textures offer an approach to directly explore and tune the topological Hall physics. The Hall signal from bimerons will be directly sensitive to the topology, enabling the electrical readout of the topological winding number and leading to new possibilities for designing magnetic-topology-based technology, where the topology encodes the information. Our findings show that these AFM textures can be detected with accurate helicity and topological charge through a multimodal combination of surface-sensitive SEMPA imaging and MFM imaging. The fully compensated synthetic antiferromagnets host homochiral Néel bimerons that are stable at room temperature.

Furthermore, the spin textures are found to cluster, implying potential energetic interactions among multiple neighbors. This observation raises the intriguing prospect of intricate interactions beyond next-nearest neighbors in these systems including potential meron-antimeron lattices that can be envisaged to be nucleated by current- or optical pulses. Such complexities could significantly impact the behavior and stability of these antiferromagnetic spin textures, warranting further investigation into the underlying mechanisms. In the synthetic antiferromagnet, bimerons exhibit an important advantage over previously demonstrated AFM bimerons[43,45] due to the presence of homochiral spin textures, which makes them amenable to controlled manipulation using SOTs and in turn, opens up new possibilities for designing spintronic devices in SyAFM systems. This combines the advantages of ferromagnet, such as easy detection, with those of antiferromagnets, such as the absence of long-range stray fields.

## Methods
### Material deposition
The thin-film material stacks were deposited on thermally oxidized Si/SiO$_2$ substrates employing the Singulus Rotaris magnetron sputtering tool, which provides reproducibility and sub-Angstrom thickness accuracy. DC-magnetron sputtering at a base pressure of $4 \times 10^{-8}$ mbar

was employed for the growth of the metallic layers Ta, Pt, Ir, $Co_{0.6}Fe_{0.2}B_{0.2}$(CFB), $Fe_{0.6}Co_{0.2}B_{0.2}$(FCB), and $Co_{0.8}B_{0.2}$ at room temperature. The respective deposition rates were determined with X-ray reflectivity to be 0.54, 0.91, 0.56, 0.51, 0.66, and 0.37 Ås$^{-1}$ under pure Ar flow used as sputtering gas.

## SEMPA imaging

For imaging the in-plane component of the magnetic spin texture we used a surface-sensitive technique, the SEMPA[52]. SEMPA is a powerful in-house imaging technique that uses the spin-polarized secondary electrons emitted from a magnetic material and gives a two-dimensional (2D) vector map of the IP magnetization[57]. The sensitivity of SEMPA is limited to 1−2 nm depth from the surface, which enables us to image the topological spin textures present only in the topmost magnetic layer. This unique feature of SEMPA is especially effective on synthetic antiferromagnets enabling us to investigate the formation of topological spin textures even in a fully compensated composition.

SEMPA color-coded images enable us to determine the winding number of the topological spin textures and classify them accordingly. Also, the sense of the in-plane rotation gives information about the exact helicity of these meron spin structures. We note that SEMPA images do not differentiate the OOP component of the magnetization, hence merons and antimerons of equal helicity give similar SEMPA color contrast. This prohibits in determining the topological charge solely from the IP contrast.

## Micromagnetic approach

**Analytical expression for the phase boundaries.** We explore the possible ground states of the model (1) along the lines of ref. [32]. We consider the most generic ansatz for a helix in the real space, which can be parametrized by the normal $\boldsymbol{n}$ to the plane of the helix and the helical pitch vector $\boldsymbol{q}$. The Néel order can be cast in terms of this parametrization as

$$\boldsymbol{l}(\boldsymbol{r}) = \cos(\boldsymbol{q} \cdot \boldsymbol{r})\boldsymbol{e}_1 + \sin(\boldsymbol{q} \cdot \boldsymbol{r})\boldsymbol{e}_2 + m_0\boldsymbol{n} \qquad (2)$$

where $\boldsymbol{l}(\boldsymbol{r}) = \boldsymbol{L}(\boldsymbol{r})/|\boldsymbol{L}|$ and $\{\boldsymbol{e}_1, \boldsymbol{e}_2, \boldsymbol{n}\}$ defines a local frame of reference in the spin space. Upon substituting this expression into Eq. (1), we obtain the following identity for the energy density functional:

$$\varepsilon[\boldsymbol{l}(\boldsymbol{r})] = \frac{1}{1+m_0^2}\left\{\frac{J}{2}\boldsymbol{q}^2 + D\left(q_x \sin\theta\sin\phi - q_y\sin\theta\cos\phi\right) \right.$$
$$\left. + K\left(\left[\frac{1}{2}+m_0^2\right] + \left[\frac{1}{2}-m_0^2\right]\cos^2\theta\right)\right\}. \qquad (3)$$

This functional is minimized with respect to the variables $\{\theta, \phi, \boldsymbol{q}, m_0\}$ and the different possible extrema are found. The lowest energy configuration for a given set of parameters determines the ground state. The phase boundaries separating any two possible ground states are determined by equating their corresponding energies, from which the expression of the DMI $D$ as a function of $K_{\text{eff}}$ is obtained. For instance, the phase boundary for the OOP-helical transition is parametrized by the curve $D_c^{\text{OOP}}(\delta) = \frac{4}{\pi}\sqrt{A\left[K_{\text{eff}} - \frac{H_d^2}{2\delta L^2}\right]}$.

**Micromagnetic simulations.** Micromagnetic simulations were performed using the Mumax3 software[58,59]. The following setup was implemented in the simulations leading to Fig. 4. A bilayer square geometry of lateral size 512 nm and thickness 1 nm for each of the FM layers was considered and dipolar interaction was included. The system was discretized with a mesh size of $1 \times 1 \times 1$ nm$^3$ and periodic boundary conditions along the $x$ and $y$ directions were imposed, with period equal to 16 repetitions. The material parameters are $A = 1 \times 10^{-11}$ Jm$^{-1}$ for the exchange constant, $M_s = 1.45$ MAm$^{-1}$ for the

saturation magnetization and $\alpha = 0.01$ for the Gilbert damping. The strength of the interlayer exchange coupling was chosen to be $2\delta = 0.44 \times 10^{-3}$ Jm$^{-2}$, which corresponds to the value obtained from SQUID measurements. We note that in Mumax$^3$, interlayer exchange interactions are properly accounted for by rescaling the material parameters by the thickness of the spacer (see ext_scaleExchange function)[59]. To explore the $D - K_{\text{eff}}$ phase diagram, the effective uniaxial anisotropy $K_{\text{eff}}$ and the DMI $D$ were varied in the range $[-3 \times 10^5, 3 \times 10^5]$ Jm$^{-3}$ and $[0, 2 \times 10^{-3}]$ Jm$^{-2}$, respectively. An initial meron configuration is chosen in the simulations, which is minimized to find the equilibrium configuration. The parameter space of $(D, K_{\text{eff}})$ was swept to obtain the light green shaded region in Fig. 4a.

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

## Acknowledgements

The authors thank A. Bose, A. Rajan and E. Galindez Ruales for their participation in additional experiments included in the Supplementary. This work has received funding from the European Union's Horizon 2020 research and innovation program under the Marie Skłodowska-Curie Grant Agreement No. 860060 "Magnetism and the effect of Electric Field" (MagnEFi), as well as from Synergy Grant No. 856538, project "3D-MAGiC" and the Horizon Europe Project No. 101070290 (NIMFEIA). It has also been supported by the Deutsche Forschungsgemeinschaft (DFG, German Research Foundation) - SPP 2137 Skyrmionics (project 462597720), TRR 173 – 268565370 (project A01 and A03), TRR 288 – 422213477 (project A09), project 445976410 and project 448880005, the Grant Agency of the Czech Republic grant no. 19-28375X, and the Dynamics and Topology Centre TopDyn funded by the State of Rhineland Palatinate. This work was partly supported by JSPS Kakenhi (No. 23K13655) and the DAAD. T.D. gratefully acknowledges financial support from the Canon Foundation in Europe.

## Author contributions

T.D., R.F., J.S. and M.K. supervised the study. T.D. and M.A.S. designed and prepared the stack structure. M.B. conducted SQUID magnetization measurements, MFM imaging and analyzed the data along with T.D; M.B. performed the SEMPA imaging together with R.F.; M.B. performed the XMCD-PEEM imaging and analysed the data with the support from M.F. and M.A.N.; M.B., V.K.B., and R.Z. did the micromagnetic simulations. V.K.B. and R.Z. developed the theoretical model and analyzed the phase

diagram, together with J.S.; M.B., T.D., and R.Z. drafted the manuscript with help of R.F., J.S., and M.K.; All authors discussed the results and commented on the manuscript.

## Funding

## Competing interests
The authors declare no competing interests.
