## [Peer Review File · Nature Communications]

REVIEWER COMMENTS

Reviewer #1 (Remarks to the Author):

Miao and co-workers have systematically investigated in-plane antiferromagnetic spin textures in the designed synthetic antiferromagnetic systems [bilayer $\text{Fe}_{0.6}\text{Co}_{0.2}\text{B}_{0.2}(\text{FCB})_x/\text{Co}_{0.6}\text{Fe}_{0.2}\text{B}_{0.2}(\text{CFB})_{0.9-x}$]. Based on SEMPA and MFM measurement techniques, they analyze the in-plane and out-of-plane components of the topological magnetic structure of the studied system [Stack#2b system with uncompensated magnetization], where the existence of Néel order parameter of (anti)merons and bimerons have been unveiled. Furthermore, micromagnetic simulations are also performed to clarify the relationship of the formation of meron spin textures with interlayer exchange, interlayer magnetic dipolar interactions, and interfacial DMI. Their simulated results reveal that SyAFMs with strong interlayer exchange coupling can provide a new way for exploration of stabilizing bimeron in contrast to large DMI materials with in-plane anisotropy. More interestingly, they confirm the stabilization mechanism for homochiral (anti)merons and bimerons in the compensated case originates from the DMI under comparable conditions based on helicity of (anti)merons. I think this is a very complete and interesting work and I recommend this work to publish in Nature Communications. However, before acceptance, I have some questions.

1. We know that the bimeron ($Q=wp$, $w=1$ and $p=1$) consists of meron ($Q=wp$, $w=1/2$ and $p=1$) and antimeron ($w=-1/2$, $p=-1$), where w and p denotes winding number and core polarity of spin textures. The authors have shown the distribution of merons, antimerons and bimerons in studied system, as is shown Fig.2. Previous works have reported that similar spin textures, e.g., isolated (anti)meron, bimeron, can exist in materials with weak in-plane anisotropy [e.g. antiferromagnetic topological magnetism in synthetic van der Waals antiferromagnets, etc.]. (1) I am curious about whether this kind of meron, antimeron and bimeron lattice exists, or under what conditions one can more easily obtain this structure? (2) Furthermore, I see that the distance between the meron and antimeron combinations marked black circles (single and double) in Figure 2b is different. How to tell whether a bimeron is formed based on the distance between a meron and antimeron?

2. The author find that Bloch- and Neel-type spin textures can be realized in stack#2b and stack #3 with uncompensated magnetizations, where the in-plane rotational symmetry of these structures should be destroyed. What role interlayer DMI plays in such a system?

3. It seems there are some issues in Figure 1, where the ratio of ingredients seems to be wrong. The authors give that the ferromagnetic layer consists of FCB (x) and CFB($0.9-x$), but it seems that the expression in Figure 1 is reversed. For example, x should be 0.2 not 0.7 in Figure 1(b)? And, the fully

compensated and uncompensated magnetization should be marked in Figures 1(b)-1(d), and it is more intuitive and obvious to compare with the following Figures 1(e)-(f).

Reviewer #2 (Remarks to the Author):

In their article 'Homochiral antiferromagnetic merons, antimerons and bimerons realized in synthetic antiferromagnets', the authors systematically demonstrate the material parameters required to stabilize topological vortex textures with the same helicity (homochirality) in a synthetic antiferromagnet (SyAF) heterostructure. As homochirality is a prerequisite to coherent spin-orbit torque driven motion, control of vortex helicity is an important step towards realising practical devices. While observation of topological vortices in SyAFs has been reported several times, the novelty of this research article is the controlled nucleation of homochiral structures.

The authors present analysis of their SEMPA measurements in a thorough way and corroborate their results with a theoretical model and micromagnetic simulations. As such, I would recommend this article for publication in Nature Communications, in its current form.

I have a few questions/suggestions for the authors, though the answers to which I do not deem necessary for qualifying the article for publication:

1) Fully compensated SyAF stack #1 is mentioned when discussing heterostructure composition for balancing the effective magnetic anisotropy, interfacial DM interaction, interlayer exchange, and dipolar anisotropy. The magnetometry measurements are presented in Figure 1b). Did the authors carry out comparable SEMPA/MFM measurements on this stack? Did they determine the size of the out-of-plane anisotropy in this stack? Is K_{eff} too large to stabilize vortices (either skyrmions or merons), based on the model presented?

2) The SEMPA images of stack #2a presented in Figure 5a) contains several regions with unmarked vortices – namely the chirality twists along the 180degree domain wall in the top left quadrant. The helicity of these unmarked vortices appears to me to be $\pi/2$ or $3\pi/2$. Including these in the statistics shown in 5b), does the bimodal distribution around 0 and π persist?

3) This opens up a question about in-plane anisotropy. As I understand it, Keff primarily consists of an out-of-plane anisotropy. Does including a large in-plane anisotropy drastically change the model? Could an in-plane anisotropy be another way of controlling vortex helicity?

4) The XMCD-PEEM images in supplementary Figure S3b) and c) show the in-plane spin orientation of adjacent ferromagnetic layers in the heterostructure for a single orientation of the sample/x-ray polarisation. Did the authors image the same area with a different geometry of x-ray polarisation with respect to the crystalline axes? This would give more insight to the variation of the spins at the highlighted textures.

- Could the authors add a contrast wheel to make it clear which axes the spins are oriented along?

5) Finally, these heterostructures appear suitable for electrical transport – the platinum layer, which induces the DM interaction, would also be a good spin-injector to the ferromagnetic layers. Have the authors attempted to drive the vortices with electrical currents? If so, what are the results? If not, what are the limitations?

Reviewer #3 (Remarks to the Author):

Bhukta et al. have experimentally observed homochiral antiferromagnetic merons, antimerons, and bimerons by combining MFM, SEMPA, and element-specific XMCD-PEEM techniques. In general I think this work is very interesting and original. I recommend its publication after considering the following comments.

1. The authors claim to provide multimodal vector imaging of the three-dimensional Néel order parameters; however, this essentially involves vector operations of the in-plane (IP) and out-of-plane (OOP) spin components. The authors employ a multi-layer stack, as shown in Figure 1a, but switch to a bilayer model in Figures 2c, d, and e. It is presumed that the observed topological spin textures should be on the top surface since two surface-sensitive tools are used. Nevertheless, the potential influence from the other magnetic layers should also be addressed. The authors should provide a clear explanation of this, and it might be worth exploring whether a bilayer stack could be effective.

2. The chirality cannot be definitively confirmed through MFM alone. The authors should elucidate how they confirmed the chirality of the out-of-plane (OOP) spin component. It's possible that a line-sweeping technique with PEEM might be necessary.

3. In Figure 2(c), the in-plane (IP) magnetization vector of the uppermost FMA layer is depicted. However, it is expected that an antiparallel texture would be present in the FMB layer due to AFM RKKY coupling. This assumption is based on a bilayer model, but the stack used here may produce a three-dimensional spin texture. The authors could consider increasing the number of stacks from 3 to 14. This could reveal whether the dipolar interaction starts to outweigh the AFM exchange coupling, as indicated by the OOP M-H loop. In other words, AFM exchange coupling may no longer dominate within the stack, requiring additional evidence for the spin texture in the FMB layer.

Response to the Referees

We would like to express our gratitude to the reviewers for their thorough examination of our manuscript and their valuable suggestions. We are pleased to receive their positive feedback regarding our work, and we have diligently implemented their comments and incorporated their suggestions into the revised manuscript. Below, you will find our responses to their feedback in detail.

Reviewer #1 (Remarks to the Author):

Miao and co-workers have systematically investigated in-plane antiferromagnetic spin textures in the designed synthetic antiferromagnetic systems [bilayer $\text{Fe}_{0.6}\text{Co}_{0.2}\text{B}_{0.2}(\text{FCB})_x/\text{Co}_{0.6}\text{Fe}_{0.2}\text{B}_{0.2}(\text{CFB})_{0.9-x}$]. Based on SEMPA and MFM measurement techniques, they analyze the in-plane and out-of-plane components of the topological magnetic structure of the studied system [Stack#2b system with uncompensated magnetization], where the existence of Néel order parameter of (anti)merons and bimerons have been unveiled. Furthermore, micromagnetic simulations are also performed to clarify the relationship of the formation of meron spin textures with interlayer exchange, interlayer magnetic dipolar interactions, and interfacial DMI. Their simulated results reveal that SyAFMs with strong interlayer exchange coupling can provide a new way for exploration of stabilizing bimeron in contrast to large DMI materials with in-plane anisotropy. More interestingly, they confirm the stabilization mechanism for homochiral (anti)merons and bimerons in the compensated case originates from the DMI under comparable conditions based on helicity of (anti)merons. I think this is a very complete and interesting work and I recommend this work to publish in Nature Communications. However, before acceptance, I have some questions.

Our reply:

We are delighted to acknowledge the reviewer's recognition of the significance of our research and their support for its publication in Nature Communications. Additionally, we are grateful for the valuable suggestion provided by the reviewer, which will further enhance the quality of our work.

Reviewer's comment

1. 1: We know that the bimeron ($Q=wp$, $w=1$ and $p=1$) consists of meron ($Q=wp$, $w=1/2$ and $p=1$) and antimeron ($w=-1/2$, $p=-1$), where w and p denotes winding number and core polarity of spin textures. The authors have shown the distribution of merons, antimerons and bimerons in studied system, as is shown Fig.2. Previous works have reported that similar spin textures, e.g., isolated (anti)meron, bimeron, can exist in materials with weak in-plane anisotropy [e.g., antiferromagnetic topological magnetism in synthetic van der Waals antiferromagnets, etc.].

Our reply:

We thank the referee for this valid comment. In response, we have refined the manuscript's introduction to incorporate the concept of isolated merons and antimerons existing in selected materials. The revised introduction from line 21-24 now reads as follows:

Line 21-24

“Topological spin textures in in-plane magnets, such as merons²¹, antimerons²², and bimerons^{23–26} have recently been explored, by virtue of their richer (current-induced) dynamics compared to skyrmions²⁷ and the stackability, a property that allows for denser quasi-one-dimensional racetracks in three dimensions, resulting in higher storage density.”

Instead of the previous text:

“Topological spin textures in in-plane magnets, such as merons, and bimerons²¹⁻²⁴ are recently being explored, by virtue of their richer current-induced dynamics compared to skyrmions²⁵ and the stackability property that allows for denser quasi-one-dimensional racetracks in three dimensions, resulting in higher storage density.”

[Added References]

21. Cui, Q., Zhu, Y., Liang, J., Cui, P. & Yang, H. Antiferromagnetic topological magnetism in synthetic van der waals antiferromagnets. Phys. Rev. B 107, 064422 (2023)

22. Ghosh, S., Blügel, S. & Mokrousov, Y. Ultrafast optical generation of antiferromagnetic meron-antimeron pairs with conservation of topological charge. Phys. Rev. Res. 5, L022007 (2023).

Reviewer’s comment

2. (1) I am curious about whether this kind of meron, antimeron and bimeron lattice exists, or under what conditions one can more easily obtain this structure?

Figure R1: Comparison of an isolated skyrmion with topological charge (Q) equal to 1 (a) and an isolated meron with $Q = 1/2$ (b). The spin structure along the indicated grey line profile is replicated above the

2D map for enhanced clarity. Color coordinates are provided on the left side of each image for reference. (c) an example of a meron-antimeron lattice.

Our reply:

An important difference between isolated skyrmions and isolated merons is based on the absolute value of the total spin rotation across these spin textures. Skyrmions exhibit a 2π rotation of the magnetization from left to right, whereas merons display only a π rotation, as illustrated in Figure R1(a, b). When creating a lattice of skyrmions, it is straightforward to align them side by side to form a lattice. However, with merons, the situation is different. In the example of Figure R1 (b), the meron begins with a green arrow on the left side and ends with a violet arrow on the right. Therefore, alternating merons and (anti)merons are required for a periodic rotation (an example of meron-antimeron lattice is shown in figure R1(c)). If the merons and antimerons are of opposite core polarities, a bimeron lattice can be formed; otherwise, this will result in a trivial meron-antimeron lattice.

Theoretical predictions of meron-antimeron lattices have been reported in literature, e.g., in the works of Kharkov, *et al.* Phys. Rev. Lett. **119**, 207201 (2017) and Hayami, *et al.* Phys. Rev. B **104**, 094425 (2021), along with experimental observations in Yu, *et al.* Nature **564**, 95 (2018). Additionally, the transition from a square meron-antimeron lattice to a hexagonal skyrmion lattice was observed by applying magnetic fields in Yu, *et al.* Nature **564**, 95 (2018).

While our specific study did not observe meron/antimeron lattices, it is worth mentioning that there are promising predictions regarding the potential generation of these structures through the application of ultrafast laser pulses, as discussed in Ghosh *et al.*, Phys. Rev. Res. **5**, L022007 (2023). In the future, additional experimental work that goes beyond the scope of our current manuscript is required to study the controlled nucleation the meron-antimeron lattice using both laser pulses and bipolar pulses, an approach that we previously used for skyrmions in Lemesh *et al.* Adv. Mater. **30**, 1805461 (2018). However, the necessary tasks exceed the scope of the present study and will have to be reserved for a future work. We have added this in the discussion section of the manuscript which reads now as follow in line 257-259

“This observation raises the intriguing prospect of intricate interactions beyond next-nearest neighbours in these systems including potential meron-antimeron lattices that can be envisaged to be nucleated by current- or optical pulses.”

From

“This observation raises the intriguing prospect of intricate interactions beyond next-nearest neighbours in these systems.”

Reviewer’s comment

1. (2) Furthermore, I see that the distance between the meron and antimeron combinations marked black circles (single and double) in Figure 2b is different. How to tell whether a bimeron is formed based on the distance between a meron and antimeron?

Our reply:

We thank the referee for bringing up this point and we fully agree with this statement. There is a noticeable variation in the separation of the marked meron-antimeron pairs in Fig 2b). It is possibly due to spatial fluctuations in magnetic properties, or it could be related to long-range interactions within the full neighbouring meron environment mediated by the in-plane magnetization. To quantify this variation,

we conducted the statistical analysis shown in Fig. 3c, where we clearly see a peak, which we attribute to the intrinsic size of the isolated bimeron. This distance can be reproduced in simulations with the material parameters from the experiment assuming reasonable DMI strength.

A bimeron, as defined in our study, is a pair formed by a meron and an antimeron with opposite core polarities. It is worth noting that when meron-antimeron pairs share the same core polarity, they compensate each other's net topological charge, resulting overall in topologically trivial structures with $Q_{net} = 0$.

Figure R2: Structure of a single bimeron from the images shown in Figure 2 of the manuscript. **(a)** SEMPA image showing the IP spin components of the sublattice-A FM layer, or, equivalently, the IP component of the Néel order of the meron texture in the stack #2b. **(b)** MFM image showing the OOP spin component of the sublattice-A FM layer, that is the OOP component of the Néel order of the meron structure in the same area. Dark brown and white MFM contrasts indicate the upward and the downward direction, respectively. The color map for the SEMPA image is shown on the right side **(c, d)** Zoomed-in view on the marked region from R2(a) and R2(b) showing a single bimeron. **(e)**

Micromagnetic simulation results giving the spin structures of a bimeron. The color map is consistent with the one used for the SEMPA image, displayed on the right side. **(f)** Simulated MFM phase contrast of the bimeron.

In our work, we employed multimodal 3D-vector imaging within the same region of the thin film as shown in Figure 2(a) and (b) in the manuscript. Figure R2 gives a more detailed structural analysis of a single bimeron. While Figures R2(a) and R2(b) reproduce Figures 2(a) and (b) of the manuscript, Figures R2(c) and (d) show a zoom into the marked area around one of the identified bimerons. From the information in Figure R2(c), we can identify the winding number, with the black single circle marking the meron ($w = 1/2$) and the black double circle marking the antimeron ($w = -1/2$). To ensure that these entities do not mutually cancel their topological charges when combined, we examined the identical zoomed-in area of the Magnetic Force Microscopy (MFM) phase image in Figure R2(d). Here, we observed that the phase contrast of both structures is inverted, indicating topological charges of $Q = \text{pW} = -1/2$ for both meron and antimeron. When these structures are combined, they exhibit non-zero topological charge of $Q = -1$. If these structures had the same core polarity, their MFM phase offset would exhibit the same polarity, rendering their combination trivial.

To further enhance our understanding, we have employed micromagnetic simulation to predict the in-plane magnetization components and simulate the MFM phase which denotes the OOP component a bimeron using Mumax³, and we compare these results with the experimental images, as shown in Figures R2(e) and R2(f). This comparison shows close similarity between the experimentally obtained IP images from SEMPA and OOP MFM phase contrast. Moreover, we assessed the separation between merons and antimerons, as discussed in Figure 4(d) and in Supplementary Figure S8(a).

As all materials parameters of our system but the DMI are known, we performed a series of simulations (as shown in Figure S8(a) of the supplementary information) with the DMI varied within a reasonable range of 0.14 mJm^{-2} to 0.34 mJm^{-2} . Increasing the DMI increases the separation up to 200 nm for a DMI value 0.34 mJm^{-2} . Considering the aforementioned observations and the agreement of the meron-antimeron pair separation in Figure 2(b) with the scale predicted by our simulations, we assertively and conclusively identify these pairs as bimerons.

Reviewer's comment

4. The author find that Bloch- and Neel-type spin textures can be realized in stack#2b and stack #3 with uncompensated magnetizations, where the in-plane rotational symmetry of these structures should be destroyed. What role interlayer DMI plays in such a system?

Our reply:

We thank the referee for this comment. In the stack #2b and stack #3 we have indeed uncompensated magnetization. However, uncompensated moments in the context of SyAFM means the imbalance between the saturation magnetization of the (A, B) magnetic sublattices and does not break the in-plane rotational symmetry (this is in contrast to the terminology used in exchange-bias systems, where in-plane uncompensated moments at the interface will break the in-plane rotational symmetry). Interlayer DMI in a synthetic antiferromagnet can occur for instance due to a columnar growth or current-induced lateral symmetry breaking at the interface, as shown in Han *et al*, Nature Materials **18**, 703 (2019) or Kammerbauer *et al*, Nano Lett. **23**, 7070 (2023). By introducing interlayer DMI in the system having the form of $D_{\text{inter. } z} (m_i \times m_{i+1})$, in-plane rotation symmetry can be broken. This leads to the stabilization of chiral Bloch spin textures, as discussed in Pollard *et al.*, Phys. Rev. Lett. **125**, 227203 (2020). The film systems studied here have been deposited during constant sample rotation and therefore do not show interlayer DMI.

Reviewer's comment

5. It seems there are some issues in Figure 1, where the ratio of ingredients seems to be wrong. The authors give that the ferromagnetic layer consists of FCB (x) and CFB (0.9-x), but it seems that the expression in Figure 1 is reversed. For example, x should be 0.2 not 0.7 in Figure 1(b)? And, the fully compensated and uncompensated magnetization should be marked in Figures 1(b)-1(d), and it is more intuitive and obvious to compare with the following Figures 1(e)-(f).

Our reply:

We thank the reviewer for his thorough reading. We have corrected the mix-up, in the manuscript lines 84-86, which read now:

“We have kept the FM layers as thin as possible to maximize the interlayer exchange coupling ($d_{FM} = 0.9$ nm) as well as optimized the ratio between the CFB (x nm) and FCB (0.9 – x nm) thicknesses to be in the vicinity of the spin reorientation transition (namely, to obtain a vanishing effective anisotropy $K_{eff} = K_u + K_d$).”

Instead of the previous text

“We have kept the FM layers as thin as possible to maximize the interlayer exchange coupling ($d_{FM} = 0.9$ nm) as well as optimized the ratio between the FCB (x nm) and CFB(0.9 – x nm) thicknesses to be in the vicinity of the spin reorientation transition (namely, to obtain a vanishing effective anisotropy $K_{eff} = K_u + K_d$).”

We have indicated that stacks #1 and #2a are fully compensated in Figure 1(b) and (c), and we have added uncompensation marks in Figure 1(d) as well as modified the figure 1 caption. The modified Figure 1 now appears as follows.

Whereas the previous figure 1 is shown below.

Reviewer's comment

Reviewer #2 (Remarks to the Author):

In their article 'Homochiral antiferromagnetic merons, antimerons and bimerons realized in synthetic antiferromagnets', the authors systematically demonstrate the material parameters required to stabilize topological vortex textures with the same helicity (homochirality) in a synthetic antiferromagnet (SyAF) heterostructure. As homochirality is a prerequisite to coherent spin-orbit torque driven motion, control of vortex helicity is an important step towards realising practical devices. While observation of topological vortices in SyAFs has been reported several times, the novelty of this research article is the controlled nucleation of homochiral structures. The authors present analysis of their SEMPA measurements in a thorough way and corroborate their results with a theoretical model and micromagnetic simulations. As such, I would recommend this article for publication in Nature Communications, in its current form.

Our reply:

We are delighted to note that the reviewer has acknowledged the originality and significance of our research and has expressed their endorsement for the publication of our manuscript in Nature Communications.

Reviewer's comment

I have a few questions/suggestions for the authors, though the answers to which I do not deem necessary for qualifying the article for publication:

1) Fully compensated SyAF stack #1 is mentioned when discussing heterostructure composition for balancing the effective magnetic anisotropy, interfacial DM interaction, interlayer exchange, and dipolar anisotropy. The magnetometry measurements are presented in Figure 1b). Did the authors carry out comparable SEMPA/MFM measurements on this stack? Did they determine the size of the out-of-plane anisotropy in this stack? Is K_{eff} too large to stabilize vortices (either skyrmions or merons), based on the model presented?

Our reply:

We thank the referee for raising this point. In the case of Stack #1, as shown in Figure 1(b), we have determined an effective anisotropy of 0.2 MJm^{-2} , indicating a positive anisotropy and thereby leading us to anticipate the presence of an out-of-plane (OOP) multidomain state in the sample. Unfortunately, due to the stack's complete compensation, Magnetic Force Microscopy (MFM) imaging was not feasible. To overcome this challenge, we employed surface-sensitive Scanning Electron Microscopy with Polarization Analysis (SEMPA), which enabled successful imaging of the IP domain walls of Stack #1. We have included the horizontal and vertical components of the SEMPA images in Figure R3(a) and (b) for your reference.

Figure R3: SEMPA images of Stack #1. (a) Horizontal and b) vertical in-plane components of the surface magnetization. The direction of magnetization is indicated by the grayscale contrast, as displayed on the double arrows.

Reviewer's comment

2) The SEMPA images of stack #2a presented in Figure 5a) contains several regions with unmarked vortices – namely the chirality twists along the 180degree domain wall in the top left quadrant. The helicity of these unmarked vortices appears to me to be $\pi/2$ or $3\pi/2$. Including these in the statistics shown in 5b), does the bimodal distribution around 0 and π persist?

Our reply:

The statistics presented in Figure 5(b) of the manuscript include the helicity information of all merons in the SEMPA images given in Figure 5(a). In the original Figure 5(a) we specifically highlighted only merons with helicity π and 0 for clarity. The referee is correct that there are two instances within Figure 5(a) where $\pi/2$ and $3\pi/2$ merons are observed (now denoted by a triangle and star in Figure 5(a)). These specific occurrences contribute to the observed minor counts around $\pi/2$ and $3\pi/2$ in Figure 5(b). To enhance clarity, we have incorporated the representation of these two merons (identified by a triangle and star shape) in the upper half of Figure 5(a) in the main manuscript. The revised Figure 5 is presented below.

Reviewer's comment

3) This opens up a question about in-plane anisotropy. As I understand it, Keff primarily consists of an out-of-plane anisotropy. Does including a large in-plane anisotropy drastically change the model? Could an in-plane anisotropy be another way of controlling vortex helicity?

Our reply:

During the deposition of our SyAFM systems, we rotate the substrate at 60 rotations per minute (RPM), which ensures a laterally isotropic deposition of the thin film. This, in turn, ensures a vanishing in-plane anisotropy, and hence, the system shows a pure easy-plane behaviour. In response to the reviewer's suggestion, we conducted in-plane hysteresis loop measurements for stack #2a along two perpendicular in-plane directions, as illustrated in Figure R4 using SQUID magnetometry. The curves at 0 degrees and 90 degrees represent the in-plane hysteresis loops along these directions. The overlapping nature of

both loops indicates that the system does not prefer any specific in-plane direction, thereby possessing an easy-plane anisotropy.

Figure R4: Magnetic hysteresis loops (IP) of stack #2a measured along two perpendicular in-plane directions using SQUID magnetometry.

We agree that if we deliberately introduced a substantial in-plane anisotropy along a defined axis, this could significantly alter the observed behavior. Under these conditions, the interfacial Dzyaloshinskii-Moriya interaction (DMI) alone may not suffice to stabilize merons within the thin film. In the studies conducted by Zarzuela *et al.* Physical Review B **101**, 054405 (2020) and Börge *et al.* Physical Review B **99**, 060407(R) (2019) it has been demonstrated that the prerequisites for DMI to stabilize merons are different in such cases. In the scenario of an interfacial DMI (isotropic case), the DMI vector lies in the film plane. However, in the scenarios detailed by Zarzuela *et al.* in Phys. Rev. B. **101**, 054405 (2020) and Börge *et al.* in Phys. Rev. B. **99**, 060407(R) (2019), to stabilize bimerons in thin films with in-plane anisotropy, the required symmetry entails a form of DMI with a non-zero out-of-plane component.

A bimeron exhibits distinctive characteristics beyond core polarity and helicity. One potential approach for its characterization is to define γ as the angle between the connecting line joining the centers of merons and antimerons relative to the x-axis, and α as the orientation of the net magnetization with respect to the x-axis, aligning parallel to its surroundings. Theoretically, it is feasible to manipulate the angle γ that connects meron and antimeron centers by altering the direction of in-plane anisotropy. Importantly, such manipulation does not impact the helicity of the merons.

Reviewer's comment

4) The XMCD-PEEM images in supplementary Figure S3b) and c) show the in-plane spin orientation of adjacent ferromagnetic layers in the heterostructure for a single orientation of the sample/x-ray polarisation. Did the authors image the same area with a different geometry of x-ray polarisation with respect to the crystalline axes? This would give more insight to the variation of the spins at the highlighted textures.

- Could the authors add a contrast wheel to make it clear which axes the spins are oriented along?

Our reply:

For the Figures S3(b) and S3(c), we imaged the same area but with different photon energies. Magnetic contrast is obtained by calculating the asymmetry between images taken with right and left-circularly polarized radiation. X-ray absorption (XAS) spectra around the Fe and Co L₃ edges find the maxima at 707.0 eV and 777.6 eV, respectively. We have added this to the section 3 of the supplementary information which reads as follows.

“Figures S3 b,c show XMCD-PEEM images of the SyAFM stack at room temperature of the same sample area but with the X-ray energy resonantly tuned to either the Co or the Fe L₃ absorption edge.”

From previously:

“Figure S3 b,c shows XMCD-PEEM images of a SyAFM stack imaged at room temperature”

We have also added contrast arrows to Figures S3(b) and (c) indicating the in-plane projection of the helicity vector of the incoming circularly polarized X-rays. The new figure looks like this

From

5) Finally, these heterostructures appear suitable for electrical transport – the platinum layer, which induces the DM interaction, would also be a good spin-injector to the ferromagnetic layers. Have the

authors attempted to drive the vortices with electrical currents? If so, what are the results? If not, what are the limitations?

Our reply:

We thank the reviewers for his/her suggestion and fully agree that our multilayers are potentially well-suited to investigate the current-induced dynamics of merons and antimerons. However, for this study we explored the stabilization mechanisms of the (anti)merons in the SyAFM stack. We would like to emphasize that this is our next goal to study the dynamics of the (anti)merons. However, access to synchrotron facilities imposes some time limitations on performing time-resolved dynamics measurements. So, such measurements and studies fall outside the immediate scope of this work but will be reserved for future research.

Reviewer #3 (Remarks to the Author):

Bhukta et al. have experimentally observed homochiral antiferromagnetic merons, antimerons, and bimerons by combining MFM, SEMPA, and element-specific XMCD-PEEM techniques. In general I think this work is very interesting and original. I recommend its publication after considering the following comments.

Our reply:

We wish to express our gratitude to the reviewer for their meticulous and thoughtful examination of our manuscript.

Reviewer's comment:

1. The authors claim to provide multimodal vector imaging of the three-dimensional Néel order parameters; however, this essentially involves vector operations of the in-plane (IP) and out-of-plane (OOP) spin components. The authors employ a multi-layer stack, as shown in Figure 1a, but switch to a bilayer model in Figures 2c, d, and e. It is presumed that the observed topological spin textures should be on the top surface since two surface-sensitive tools are used. Nevertheless, the potential influence from the other magnetic layers should also be addressed. The authors should provide a clear explanation of this, and it might be worth exploring whether a bilayer stack could be effective.

Our reply:

We thank the referee for this comment. In our study, we employ a multilayer SyAFM stack, as depicted in Figure 1(a). We have chosen to utilize multiple repetitions for two primary reasons. The first reason is to minimize the thermal diffusion of the topological spin textures (even for using materials with small pinning strength). Previous experiments conducted by our group have demonstrated that skyrmions within bilayer SyAFMs exhibit significant thermally induced diffusive motion, as reported in Dohi *et al.*, Nat. Commun. **14**, 5424 (2023). For the purpose of performing 3D vector imaging of the magnetization, we have employed a combination of the MFM and SEMPA techniques. However, the time to transition from one setup to another is of the order of days due to the stringent ultra-high vacuum requirements. Moreover, SEMPA is an inherently slow imaging technique, with each single in-plane 2D magnetization vector image taking 3-4 hours to acquire. To ensure precise determination of the chirality and helicity of the domains and spin textures, it is imperative that these spin textures remain unaffected by thermal diffusion over time scales of days.

The second reason is to maximize the effective AFM exchange field, as demonstrated in Hellwig et al, J. Magn. Magn. Mater. **319**, 13 (2007). In line with this, our prior pre-characterization experiment on stack #1 has indicated that increasing the number of repetitions in the SyAFM stack results in an augmentation of the effective AFM exchange field (H_{sat}), as depicted in Figure R5(a). The strong interlayer exchange coupling significantly reduces the critical DMI required to stabilize bimerons, as described on page 11 of the main manuscript, lines 203-210:

“For a strongly AFM-coupled SyAFM (i.e, large δ), in the vicinity of the SRT point, only a very small DMI is needed to induce the phase transition from the uniform IP configuration to a helical phase (denoted by the green and bronze areas). The low D_c stems from the fact that the only contribution to the effective anisotropy at the SRT point comes from the interlayer magnetic dipolar field. We observe that the critical value D_c is proportional to $1/\sqrt{\delta}$, leading to an increase in D_c for weak AFM interlayer couplings, as shown in Fig. 4(b) with $1 \times 10^{-5} \text{ Jm}^{-2}$, and Fig. 4(c) with $1 \times 10^{-6} \text{ Jm}^{-2}$. Achieving a larger DMI for an easy-plane system can be exceedingly challenging experimentally; consequently, systems with strong AFM interlayer coupling offer a more viable route for stabilizing bimerons.”

Figure R5(a): Magnetic hysteresis loop as a function of the number of repetitions of the SyAFM stack#1

Furthermore, we conducted micromagnetic simulations for stack #2a, varying the number of repetitions in the SyAFM layer. Our results confirm that the saturation field does indeed increase with an increasing number of layers, as depicted in Figure R6.

Figure R6(a,b): Micromagnetic simulation of magnetic hysteresis loops for stack #2a as a function of the number of repetitions of the SyAFM for (a) IP hysteresis loop (b) OOP hysteresis loop.

Potential Influence from the other layers

Regarding the comment on the magnetic modelling of the multi-layer stack, we have not switched to a bilayer model. The magnetic layers within the SyAFM stacks discussed in Figure 1 are labelled by FM_A and FM_B according to the emergent bipartite (magnetic) lattice structure, which is ascertained by the strong AFM coupling. The existence of layers deeper in the stack with a significantly weaker coupling can be excluded from the observed hysteresis curves, as discussed in the following: We conducted micromagnetic simulations to scrutinize the potential impact of insufficient coupling strength. Employing the experimental parameters for 'stack#2,' we systematically performed these simulations. We focused on the computation of out-of-plane (OOP) and in-plane (IP) hysteresis loops for a SyAFM comprising 28 layers, all of which were fully antiferromagnetically (AFM) coupled. These results are depicted by the red curves in Figure R7(a) and R7(b) respectively. Conversely, the blue curve corresponds to a configuration where 26 layers are antiferromagnetically coupled, and 2 layers are ferromagnetically (FM) coupled with equal coupling strength, establishing an effective uncompensated SyAFM. This uncompensated character is visually evident in the hysteresis loop.

Figure R7: Micromagnetic simulation of magnetic hysteresis loops as influenced by the presence of ferromagnetically coupled layers within the stack. (a) In-plane hysteresis loops (b) Out-of-plane hysteresis loops

Furthermore, as we incrementally augmented the number of FM-coupled layers, we observed a commensurate increase in the degree of effective uncompensation within the stack. This observation serves as an indicator that the AFM coupling in our stack exhibits robustness and uniformity throughout the layers, preventing the occurrence of effective uncompensation within the hysteresis loop.

We have changed the wording in the caption of Fig 2 to prevent a potential misunderstanding:

Panel (c) shows the spin textures of all FM_A and all FM_B layers in the stack for an AFM Néel meron having $\gamma = 0$ and $Q = \frac{1}{2}$.

2. The chirality cannot be definitively confirmed through MFM alone. The authors should elucidate how they confirmed the chirality of the out-of-plane (OOP) spin component. It's possible that a line-sweeping technique with PEEM might be necessary.

In our work, we have employed consecutive multimodal 3D-vector imaging within the same region of the thin film as shown in figure 2(a) and (b) in the manuscript. Figure R2 presents our detailed magnetic structural analysis, wherein R2(a) and R2(b) correspond to regions extracted from Figure 2(a) and (b) of the manuscript and were magnified to focus on the marked bimeron structures in Figure R2(c) and R2(d). Within Figure R2(c), we can distinguish the winding number, with the black single circle representing merons ($w = \frac{1}{2}$) and the black double circle representing antimerons ($w = -\frac{1}{2}$). To ensure that these entities do not mutually cancel their topological charges when combined, we examined the zoomed-in image of the Magnetic Force Microscopy (MFM) phase in the same sample area in Figure R2(d). Here, we observed that the phase contrast of both structures is inverted, signifying topological charges of $Q = pW = -1/2$ for both merons and antimerons. When these structures are combined, they exhibit an integral topological charge. It is important to note that if these structures have the same core polarity, their MFM phase offset would exhibit the same polarity, rendering their combination trivial. Hence using the combination of both SEMPA imaging and MFM, we have confirmed the chirality of the topological spin textures. Line-profile analysis in XMCD-PEEM only yields the projection of the magnetization onto the incoming-beam axis, mixing one in-plane component with the out-of-plane one. While this may be sufficient to extract information on topology in some cases, our approach gives access to the full 3D vector field of the magnetization and thus unambiguously yields the topology of the observed structures.

3. In Figure 2(c), the in-plane (IP) magnetization vector of the uppermost FMA layer is depicted. However, it is expected that an antiparallel texture would be present in the FMB layer due to AFM RKKY coupling. This assumption is based on a bilayer model, but the stack used here may produce a three-dimensional spin texture. The authors could consider increasing the number of stacks from 3 to 14. This could reveal whether the dipolar interaction starts to outweigh the AFM exchange coupling, as indicated by the OOP M-H loop. In other words, AFM exchange coupling may no longer dominate within the stack, requiring additional evidence for the spin texture in the FMB layer.

We would like to express our appreciation to the reviewer for their valuable comment. As mentioned in their observation, we anticipate the existence of an antiparallel texture within the FM_B layer, attributed to the AFM exchange coupling. To offer direct visualization of this phenomenon, we have conducted XMCD-PEEM imaging on stack #4, as elaborated in supplementary section 2 and as illustrated in supplementary figure S3. This section is detailed as follows.

Figure S3. XMCD-PEEM images of stack #4 at Co and Fe edges (a) Sample structure for stack #4. XMCD-PEEM images of a meron pattern in the SyAFM (b) Co and c) Fe L_3 edge contrast of the same surface area. The IP-sensitive direction is along the horizontal. White circles mark meron spin textures in the top layers of the SyAFM. The observation of an inverted contrast between the Co-rich top and the Fe-rich second FM layer confirms AFM coupling between these layers.

“We performed XMCD-PEEM experiments on stack #4 using its element specificity to prove the AFM coupling between the FM layers. XMCD-PEEM images are acquired at the ALBA synchrotron (BL-24 Circe) facility using PEEM, where the beam is incoming at an angle of 16° with respect to the sample surface. Magnetic contrast is obtained by calculating the asymmetry between images taken with right and left-circularly polarized radiation. X-ray absorption (XAS) spectra around the Fe and Co L_3 edges find the maxima at 707.0 eV and 777.6 eV, respectively. Stack #4 has an odd number of layers (29 layers) and is shown in figure S3a. We note that this SyAFM is purposefully designed to have an additional FM (CoB) on the top to see a clear contrast at both Co and Fe edges, representing consecutive FM layers. Figures S3 b,c show XMCD-PEEM images of the SyAFM stack at room temperature of the same sample area but resonantly tuned to either the Co or Fe L_3 edge. Panels b) and c) show the magnetic contrast at the Fe and Co edges. By imaging at the Fe edge, we are able to isolate the magnetic contrast from the second magnetic layer, as there is no Fe present in the top one. The centers of the observed meron spin textures are marked with white circles. In contrast, the magnetic contrast at the Co edge originates predominately from the topmost FM layer, as the second layer has effectively less Co. We find that the (anti)merons visible in this top layer are located exactly at the same positions as in the second layer, however, the magnetic contrast is inverted. This confirms the AFM coupling between the domains, domain walls, and the centers of merons and antimerons in the SyAFM system.”

In our SyAFM stack, we have carefully optimized the thickness of the Ir layer to 0.4 nm, a choice that maximizes the strength of the antiferromagnetic (AFM) exchange coupling. Furthermore, our stack incorporates a repeated structure, specifically 14 repetitions of the SyAFM layer, which significantly enhances the stack's effective saturation field.

In our stack, we have established a robust antiferromagnetic (AFM) exchange coupling, as evidenced by our analysis of the SQUID hysteresis loop data.

In response to the reviewer's comment, we conducted micromagnetic simulations of the bimeron within the multilayer SyAFM structure, employing the parameters obtained from experiments as outlined in the main manuscript. Our micromagnetic simulations were conducted while varying the number of repetitions of the SyAFM, extending up to $Z = 24$, where Z represents the count of ferromagnetic (FM) layers that are antiferromagnetically (AFM) coupled (as shown in figure R8). Our findings revealed that the size of the bimeron remains consistent even as the number of repetitions increases, resulting in a structure resembling a bimeron tube. This stability in bimeron size suggests that the spin texture, being AFM-coupled across the FM layers, does not exhibit significant changes throughout the thickness. As a

result, it is feasible to represent these spin textures effectively in a 2D model, as discussed in the manuscript.

Figure R8: Bimeron Evolution in SyAFM with Varying Repetitions. (a) 2-layer SyAFM, (b) 4-layer SyAFM, (c) 8-layer SyAFM, and (d) 24-layer SyAFM. Here, Z represents the count of FM layers antiferromagnetically coupled.

Also, one has to consider that, in contrast to the ferromagnetic case, the build-up of dipolar energy as the thickness of the SyAFM increases is negligible as compared to the AFM coupling energy, so that 3D flux closure structures are not occurring. Since, the magnetic charges are locally compensated, only a linear quadrupole moment remains from each double layer, which falls off far more rapidly with distance. Even at small uncompensation the dipolar energies, being square in the uncompensated moments, are still much less relevant than in a ferromagnet.

Moreover, if we consider an easy-plane case, the dipolar interaction favors antiferromagnetic coupling between adjacent layers, which is distinctly different from the out-of-plane easy-axis case. So, in the easy-plane case, the dipolar interaction does not compete with interlayer AFM coupling. Thus, an additional interaction like interlayer DMI would be required to stabilize 3D spin structures.

REVIEWERS' COMMENTS

Reviewer #1 (Remarks to the Author):

I thank the authors for their detailed answer to my questions. All my concerns have been fully addressed. I recommend its publication now. One more suggestion for the authors: in the introduction, the authors discussed the importance of skyrmions, merons, bimerons etc. Actually, from readers' point of view, more discussion of DMI should be helpful to expand the scope of impact. some recent reviews may help for your discussion regarding the progress of DMI (e.g. Nature Reviews Physics 5, 43; Rev. Mod. Phys. 95, 015003)

Reviewer #2 (Remarks to the Author):

The authors of 'Homochiral antiferromagnetic merons, antimerons and bimerons realized in synthetic antiferromagnets' have thoroughly answered my questions after reading the first version of their article. The revisions made have strengthened the manuscript and supplementary information and, as such, I still recommend this article for publishing in Nature Communications.

Reviewer #3 (Remarks to the Author):

The authors have satisfactorily addressed most of the comments in the first round review. Thus it can be accepted as in its present form.

Response to the Referees

Reviewer #1 (Remarks to the Author):

I thank the authors for their detailed answer to my questions. All my concerns have been fully addressed. I recommend its publication now. One more suggestion for the authors: in the introduction, the authors discussed the importance of skyrmions, merons, bimerons etc. Actually, from readers' point of view, more discussion of DMI should be helpful to expand the scope of impact. some recent reviews may help for your discussion regarding the progress of DMI (e.g. Nature Reviews Physics 5, 43; Rev. Mod. Phys. 95, 015003)

Our reply:

“These chiral topological spin textures are primarily stabilized by the interplay between the Dzyaloshinskii–Moriya interaction (DMI)^{16,17}, the perpendicular magnetic anisotropy, and the dipolar interaction. The orientation of the DMI vector, which is determined by the way the inversion symmetry is broken, sets the preferential stabilization of diverse skyrmion types, such as Bloch^{1,2}, Néel^{18,19}, or antiskyrmions²⁰. For instance, in magnetic multilayers, the combination of intrinsic inversion symmetry breaking, and strong spin-orbit coupling provided by heavy atoms at the interface of a ferromagnetic/heavy-metal heterostructure creates an ideal platform for providing interfacial DMI²¹, which in turn stabilizes Néel-type skyrmions.”

Reviewer #2 (Remarks to the Author):

The authors of 'Homochiral antiferromagnetic merons, antimerons and bimerons realized in synthetic antiferromagnets' have thoroughly answered my questions after reading the first version of their article. The revisions made have strengthened the manuscript and supplementary information and, as such, I still recommend this article for publishing in Nature Communications.

Our reply:

We are very pleased to learn that our revisions have resolved the reviewer's concerns and the reviewer now supports the publication of our manuscript in Nature Communications

Reviewer #3 (Remarks to the Author):

The authors have satisfactorily addressed most of the comments in the first round review. Thus it can be accepted as in its present form.

Our reply:

We are delighted to hear that our manuscript revisions have successfully addressed the reviewer's concerns, and we appreciate the reviewer's support for its publication in Nature Communications.